# Rejection Improves Reliability: Training LLMs to Refuse Unknown Questions Using RL from Knowledge Feedback

**Hongshen Xu**[1], **Zichen Zhu**[1], **Situo Zhang**[1], **Da Ma**[1], **Shuai Fan**[2*], **Lu Chen**[13*], **Kai Yu**[13*]
[1]X-LANCE Lab, Department of Computer Science and Engineering
MoE Key Lab of Artificial Intelligence, AI Institute
Shanghai Jiao Tong University, Shanghai, China
[2]AISpeech Co., Ltd., Suzhou, China
[3]Suzhou Laboratory, Suzhou, China
{xuhongshen, JamesZhutheThird, chenlusz, kai.yu}@sjtu.edu.cn

## Abstract

Large Language Models (LLMs) often generate erroneous outputs, known as hallucinations, due to their limitations in discerning questions beyond their knowledge scope. While addressing hallucination has been a focal point in research, previous efforts primarily concentrate on enhancing correctness without giving due consideration to the significance of rejection mechanisms. In this paper, we conduct a comprehensive examination of the role of *rejection*, introducing the alignment goal of model reliability along with corresponding metrics. This goal requires the model to provide accurate responses while adeptly rejecting questions exceeding its knowledge boundaries, thereby minimizing hallucinations. To improve the inherent reliability of LLMs, we present a novel alignment framework called *Reinforcement Learning from Knowledge Feedback* (RLKF). RLKF leverages knowledge feedback to dynamically determine the model's knowledge boundary and trains a reliable reward model to encourage the rejection of out-of-knowledge questions. Experimental results on mathematical and question answering datasets affirm the substantial efficacy of RLKF in significantly enhancing LLM reliability.

## 1 Introduction

Large Language Models (LLMs) have exhibited strong capabilities in solving various downstream tasks through alignment techniques such as Supervised Finetuning (SFT, (Zhang et al., 2023b), (Zhang et al., 2023b)), Direct Preference Optimization (DPO, (Rafailov et al., 2024)), and Reinforcement Learning from Human Feedback (RLHF, (Stiennon et al., 2020; Ouyang et al., 2022)). Those techniques align language models with human intent, mainly to maximize the helpfulness of LLM's responses. However, maximizing helpfulness does not mean minimizing errors. A significant problem arises in that LLMs often produce outputs that, while seemingly plausible, contain factual errors (Min et al., 2023) or self-contradictions (Liu et al., 2022), which are referred to as hallucinations.

To mitigate the hallucinations, many studies focus on augmenting the knowledge of LLMs, such as curating training data (Penedo et al., 2023; Zhou et al., 2023) or employing retrieval-augmented generation (RAG, (Gao et al., 2023b;b)) during inference. Nevertheless, it is essential to acknowledge that model knowledge inherently has limitations, and even the most powerful models, such as GPT-4, are prone to experiencing hallucinations (Zhang et al., 2023a). Consequently, we posit that the fundamental nature of the hallucination problem lies in the model's *misalignment with its knowledge boundary*. Hallucinations arise when LLMs try to answer questions beyond their knowledge boundary.

---

*The corresponding authors are Lu Chen, Shuai Fan and Kai Yu.

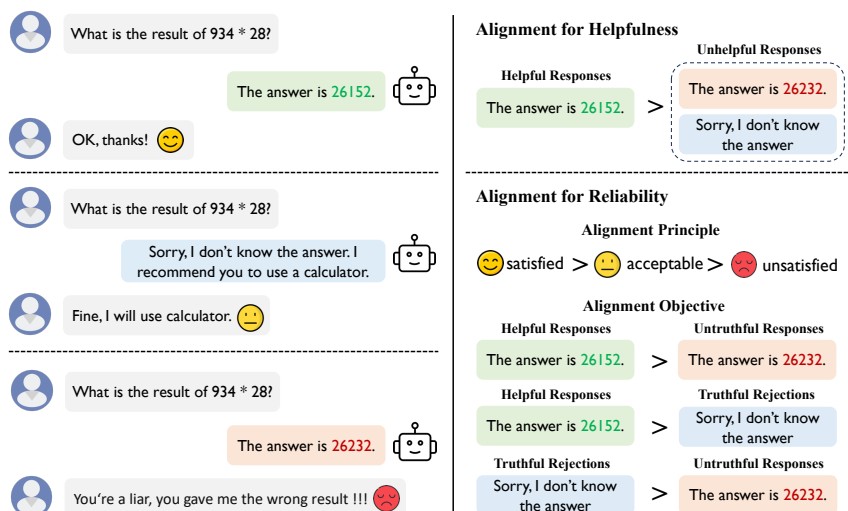

Figure 1: The user cases and alignment objectives for model reliability.

One research direction related to aligning LLM with its knowledge boundary involves alignment for honesty (Kadavath et al., 2022; Yang et al., 2023b). However, alignment for honesty presents significant challenges. On the one hand, the honesty of LLMs is hard to evaluate. Honesty can be viewed as a classification problem, where the model learns to distinguish between what it knows and what it does not. Nevertheless, the accuracy which indicates how much the model knows can differ dramatically among various models or even with different prompts (Kaddour et al., 2023). On the other hand, it is difficult to employ honesty for comparing and selecting different models consistently. Model honesty is tied to the capabilities of the model itself; a more honest model does not necessarily imply that it will provide more assistance or make fewer mistakes compared to other models.

To address the aforementioned limitations, we introduce the alignment goal of *reliability* inspired by model truthfulness (Lin et al., 2021). We define the reliability from the user's perspective. As shown in Figure 1, helpful responses (correct answers) can assist users while untruthful responses (incorrect answers) result in user loss, leading to distrust in the model. Therefore, we contend that the key to achieving a reliable system lies in ***providing as many correct answers as possible to maximize helpfulness while learning to explicitly refuse unknown questions to minimize errors.*** We apply *accuracy* and *truthfulness* to measure the model's helpfulness and errors, respectively, and further propose an overall *reliability score* to simultaneously evaluate the above two aspects.

To optimize model reliability, we propose the Reinforcement Learning from Knowledge Feedback (RLKF) training framework based on RLHF. On the one hand, most publicly available alignment data (Cui et al., 2023) often originate from multiple source models, which is impossible to be used for aligning target model with its own knowledge boundaries. On the other hand, while the preference data in RLHF is derived from the target model's outputs, it is heavily influenced by human annotators' biases. The current annotation goals based on helpfulness lead the reward model in RLHF to only learn to distinguish the helpfulness of responses, making it difficult to discern their truthfulness. Instead, RLKF automates the construction of preference data for a specific target model through knowledge feedback rather than human feedback. Knowledge feedback is primarily used to assess whether a question falls within the model's knowledge boundary. When a question falls within the model's knowledge boundary, a response is preferred over a refusal. Conversely, when a question exceeds the model's knowledge boundary, refusal is preferred over a response. The synthesized preference data is used to train a reliable reward model, which thoroughly understands the target model's knowledge boundaries and further instructs the target model on when to respond and when to refuse through the PPO algorithm. Experimental results

further demonstrate the effectiveness of our framework, which significantly improves the reliability of baseline models.

The contributions of this paper are summarized as follows:

- We introduce the alignment goal of model reliability and define several metrics to assess the reliability of LLMs.
- We propose the Reinforcement Learning from Knowledge Feedback (RLKF) alignment framework to improve LLM reliability.
- Extensive experiments are conducted to validate the effectiveness of RLKF framework.

## 2 Problem Formulation

### 2.1 LLM Alignment

With the potential risks brought by powerful LLMs, researchers have developed various alignment approaches to align LLMs with human instruction, preference, and values (Wang et al., 2024). Specifically, for the input prompt $x_i$ and alignment goal *helpfulness*, we can employ the following scoring principle to represent our alignment objective: $s(x, y_h) > s(x, y_u)$, where $y_h, y_u$ represent a helpful response and an unhelpful response, respectively. The scored response pair can be annotated either by human annotators (Ouyang et al., 2022) or a scoring model (Gao et al., 2023a) trained with human preference data. We can further utilize this comparison data to train a reward model or LLM policy, thus aligning LLMs with specific goals.

Furthermore, for a given set of $N$ inputs and LLM response $y_i$ of each input $x_i$, we can simply evaluate the helpfulness of LLM using accuracy, where helpful responses are considered correct and unhelpful responses are considered incorrect.

### 2.2 Alignment with Reliability

While many works focus on alignment for helpfulness, existing alignment goals make it hard to alleviate model-specific hallucinations. We further propose the alignment goal of model reliability from the user's perspective. We believe that *a reliable system should be aligned with user experience that provides as much assistance as possible while making as few errors as possible*.

Specifically, for the input prompt $x_i$ and alignment goal *reliability*, we employ the following scoring principle to represent our alignment objective:

$$s(x, y_c) > s(x, y_r) > s(x, y_w), \tag{1}$$

where $y_c, y_r, y_w$ represent a helpful response (the correct answer), a truthful rejection, and an untruthful response (the wrong answer), respectively. We then use *accuracy* (acc) for assessing the helpfulness, and *truthfulness* (truth) (Lin et al., 2021), representing the proportion of truthful, non-harmful responses. We also include *precision* (prec) to partially reveal the model's self-knowledge:

$$prec = \frac{N_c}{N_c + N_w}, acc = \frac{N_c}{N}, truth = \frac{N_c + N_r}{N} = 1 - \frac{N_w}{N} \tag{2}$$

where $N_c, N_r, N_w$ represent the number of correct, rejected and wrong responses, respectively. Finally, we define a comprehensive system *reliability* (rely) metric based on accuracy and truthfulness:

$$rely(\alpha) = \alpha * truth + (1 - \alpha) * acc, \tag{3}$$

where $\alpha \in [0, 1]$, and it represents the degrees of sensitivity among users towards errors. As alpha increases, reflecting greater user emphasis on system truthfulness, the model should aim to minimize errors by using refusal to respond when appropriate to meet

user expectations. Specifically, when $\alpha$ equals to answer rate (ans.), we define the overall reliability as:

$$ans. = 1 - \frac{N_r}{N}, rely = ans. * truth + (1 - ans.) * acc. \tag{4}$$

When the answer rate is low, we encourage the model not to reflexively refuse but rather to attempt to provide assistance. Conversely, when the answer rate is high, we believe the model should become more cautious to avoid errors. This metric balances the model's helpfulness and truthfulness while mitigating the risks of it becoming overly conservative or excessively aggressive.

## 3 RLKF

To better align LLMs with reliability, we propose the *Reinforcement Learning from Knowledge Feedback* (RLKF) framework. By introducing knowledge feedback, our framework trains LLMs to learn to refuse out-of-knowledge questions explicitly to achieve the alignment goal.

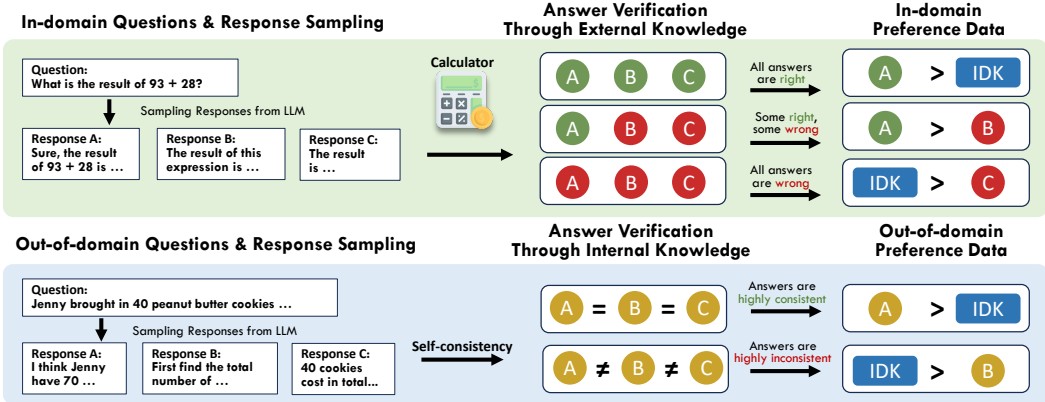

Figure 2: Reliable preference data generation pipeline. Letters with green, red, and yellow circles denote correct, incorrect, and uncertain answers, respectively. "IDK" represents "I don't know," indicating rejections.

### 3.1 High-level methodology

The proposed Reinforcement Learning from Knowledge Feedback (RLKF) framework is built upon RLHF (Ouyang et al., 2022) for reliability alignment. Instead of annotating preference pairs with human labor in RLHF, we automatically generate reliable preference pairs with knowledge feedback. The key to constructing reliable preference data lies in the insertion of rejection responses within the comparison set as defined in §2.2. Thus aligned LLM can understand the importance of avoiding errors through rejection. We describe the high-level methodology as follows:

We start with a model that is already aligned for helpfulness or harmlessness, i.e., LLAMA-2 CHAT (Touvron et al., 2023), and then apply the following three steps below:

**Step 1: Synthesizing model-specific reliable preference data through knowledge feedback.** Given the *LLM* and each input prompt $x$ in prompt dataset $\mathcal{D}$, we dynamically construct different comparison pairs according to the model responses to formulate reliable preference data *RPD*.

**Step 2: Train a reliable reward model with collected preference data.** We first train the reward model using generic helpfulness preference data *PD* to obtain *RM*. Then, we continuously train *RM* with the synthesized reliable preference data to obtain a reliable reward model *RRM*.

**Step 3: Optimize a policy against the reward model using PPO.** We use the output of the *RRM* as a scalar reward. We fine-tune the supervised policy *LLM* to optimize this reward using the PPO algorithm (Schulman et al., 2017).

## 3.2 Reliable preference data synthesizing

Compared to normal preference data that only classifies LLM responses into helpful and unhelpful classes, we categorize the LLM responses into three types: helpful, truthful, and untruthful responses, which correspond to correct answers, rejections, and incorrect answers, respectively. Besides, we also need to dynamically select two types out of three types of responses for constructing a comparison pair based on different questions. We believe that the key to selecting lies in whether the problem is within the model's knowledge boundary (whether the model can answer the question correctly). We consider two types of research settings. In the in-domain setting, we can acquire the correctness of model predictions through external knowledge, i.e., a calculator for arithmetic questions. In the out-of-domain scenario, we only have input prompts and cannot obtain golden labels of the inputs.

### 3.2.1 In-domain Reliable Preference Data

As depicted in the upper portion of Figure 2, we determine whether a question lies within the model's knowledge boundary by analyzing the distribution of correctness across multiple samplings of model responses, then construct different comparison pairs accordingly. Specifically, given the input $x$ and $N$ sampling responses $y_i, i \in [1, N]$, the comparison pair is selected as follows:

$$pair = \begin{cases} (x, y_c) > (x, y_r), & \text{if } N_c == N, \\ (x, y_c) > (x, y_w), & \text{if } N_c \in [1, N), \\ (x, y_r) > (x, y_w), & \text{if } N_c == 0, \end{cases} \tag{5}$$

where $y_c, y_r, y_w$ represents the random choice from correct, reject, and wrong responses, respectively, and $N_c$ represents the number of correct responses. In cases where the model lacks specific rejection responses, we randomly select one rejection sentence from 50 sentences that are generated by ChatGPT based on rejection templates. The templates can be found in Appendix J.

### 3.2.2 Out-of-domain Reliable Preference Data

In the out-of-domain setting, we utilize self-consistency Wang et al. (2022) as the internal model knowledge to assess whether the model possesses sufficient knowledge about the given question. As depicted in the lower portion of Figure 2, when the samplings exhibit high consistency, we align the model with preferring to provide an answer; conversely, when the samplings demonstrate low consistency, refusing is more preferred. Specifically, given the input $x$ and $N$ sampling responses $y_i, i \in [1, N]$, the comparison pair is selected as follows:

$$pair = \begin{cases} (x, y_a) > (x, y_r), & if N_s > t, \\ (x, y_r) > (x, y_a), & if N_s < t, \end{cases} \tag{6}$$

where $y_a, y_r$ represents the random choice from answered and rejected responses, respectively, $N_s$ represents the number of the most consistent answer over all answers, and $t$ represents the threshold of consistency score where we use $\lceil \frac{N}{2} \rceil$ for all the experiments.

## 3.3 Model training

**RM Training.** To train the reward model, we convert our collected pairwise reliable preference data into a binary ranking label format (i.e., chosen & unchosen) and enforce the chosen response to have a higher score than its counterpart. We used a binary ranking loss

consistent with Ouyang et al. (2022):

$$L_{ranking} = -log(\sigma(r_\theta(x, y_{chosen}) - r_\theta(x, y_{unchosen}))), \tag{7}$$

where $r_\theta(x, y)$ is the scalar score output for input $x$ and completion y with model weights $\theta$. $y_{chosen}, y_{unchosen}$ are the chosed and unchosed responses, respectively.

**Reinforcement Learning (RL).** We further train our LLM policy following the RL scheme of Stiennon et al. (2020), which uses the reward model as an estimate for the true reward function. During this phase, we seek to optimize the following objective:

$$\arg\max_\pi \mathbb{E}_{p \sim \mathcal{D}, g \sim \pi}[R(g|p) - \beta D_{KL}(\pi_\theta(g|p)||\pi_0(g|p))] \tag{8}$$

We iteratively improve the policy by sampling prompts $p$ from our dataset $\mathcal{D}$ which contains both in-domain and out-of-domain prompts and generations g from the policy $\pi$ and use the PPO algorithm and loss function to achieve this objective, the reward function also contains a penalty term for diverging from the original policy $\pi_0$.

# 4 Experiments

## 4.1 Experiment Setup

### 4.1.1 Dataset Construction

**Mathematical Dataset.** Our experiments involve two mathematical datasets: synthesized arithmetic questions and GSM8K (Cobbe et al., 2021). The arithmetic dataset consists of 14,000 samples, divided into sets of 10,000, 3,000, and 1,000 samples for training the reward model, training the policy as prompt data, and testing the final results, respectively. Additionally, we sampled 2,000 data points from the training set of GSM8K to train the reward model and 1,000 data points to optimize the LLM policy. It is important to note that we only used the input prompt of the GSM8K data points, without utilizing the annotations. This serves as an out-of-domain experimental setting.

We generate the arithmetic dataset synthetically following Liu & Low (2023). The input numbers are randomly generated, hence ensuring a very low probability of instances being duplicated. We sample from log space to ensure the numbers are equally likely to be sampled from different orders of magnitude. Following Liu & Low (2023), we use hundreds of instruction templates generated by ChatGPT, e.g., `Please help me calculate {arithmetic}.`, to diversify the question formats. Examples of templates can be found in Table 9 of Appendix I.

**Knowledge-based QA Dataset.** TriviaQA (Joshi et al., 2017) is a widely-used QA dataset that can be used to test a model's world knowledge. We selected 20,000 samples from the TriviaQA training set for training. Since the ground truth of the TriviaQA test set is not publicly available, we used the TriviaQA development set, which contains 11,313 samples, to validate our results. We used the Exact Match metric (whether the answer is exactly in the model's response) from TriviaQA paper as our measure of Accuracy, while keeping the other metrics consistent with those in mathematical datasets.

### 4.1.2 Baselines

We incorporate several baselines to benchmark the performance of our RLKF framework. All prompts used in this work are listed in Appendix G.

**No & prudent system prompt.** System prompts are commonly used to control a model's response style and personality. When the system prompt is empty, LLAMA 2-CHAT tends to respond to almost every question, losing the ability to reject unknown questions. We also use the default system prompt of LLAMA 2-CHAT introduced by Touvron et al. (2023) as the prudent system prompt. LLAMA 2-CHAT will become more cautious and reject more questions with this system prompt. These two types of prompts represent two different personalities of LLAMA 2-CHAT, serving as important baselines for this study.

**In-context Learning.** We randomly sampled 5 correctly responded (if the model can answer correctly) and 5 refused examples (if the model is unable to answer correctly) to append before the input question as the in-content learning baseline.

**Rule-based PPO.** We use a rule-based reward function to optimize the policy in in-domain prompts. The model receives a reward of 1 if it answers correctly, 0 if it refuses to answer, and -1 if it answers incorrectly. The determination of different cases is achieved through heuristic rules.

**RLHF.** We use the reward model trained only on generic helpful preference data to optimize the policy with the same training prompts as RLKF.

**SFT.** We use all 10,000 arithmetic questions from the constructed preference dataset with their chosen responses to directly fine-tune the LLM policy.

### 4.1.3 Training Details

We employ the LLAMA 2-CHAT 7B (Touvron et al., 2023) as our baseline model, which has been already aligned with human preferences. The policy model and reward model are both initialized from LLAMA 2-CHAT 7B. We use `DeepSpeed-Chat` (Yao et al., 2023) to run the whole training pipeline. As for the Reward Model (RM), we first utilized several open-source datasets (Stiennon et al., 2020; Bai et al., 2022; Ethayarajh et al., 2022) and replicated the training process following UltraRM (Cui et al., 2023) to obtain a helpful RM. Subsequently, we trained a reliable RM on the constructed reliable preference data, using a batch size of 8 for two epochs. During the RL phase, we trained for one epoch with a batch size of 1, the generation batches and PPO epochs are both equal to 1. Prompts from both in-domain and out-of-domain datasets are mixed to train the final RM. We train the SFT model for one epoch with a batch size of 32. All other training parameters were set to the default in `DeepSpeed-Chat`. We conduct all experiments using Nvidia A800 GPUs.

### 4.1.4 Evaluation Details

For evaluating the final policy, we utilize four metrics: precision, accuracy, truthfulness, and reliability. Precision is the proportion of correctly answered questions among those the model chose to answer, reflecting its self-awareness of its own capabilities. Accuracy is the proportion of correct answers among all questions, indicating how much assistance the model provides to the user. Truthfulness is the proportion of questions the model either answered correctly or refused to answer. Reliability is the dynamic weighting of accuracy and truthfulness; high reliability indicates that the model provides more help to the user and less incorrect information. The specific evaluation formulas are detailed in Section 2.2. To determine whether the policy refuses to respond and extract answers, we employ an additional answer extractor LLM (LLAMA 2-CHAT 7B with extractor prompt, see Appendix G). By comparing the extracted answers with the standard answers, we can ascertain their correctness.

## 4.2 Reliability Evaluation

**Reliability on Mathematical Tasks.** Table 1 presents the results of the final policy model after RLKF on arithmetic questions and GSM8K datasets. We can see that RLKF significantly enhances the model's reliability on both in-domain and out-of-domain datasets. The RLKF-trained model shows remarkable improvements in precision, truthfulness, and reliability. The increase in precision reflects the model's increase in self-knowledge, while the improvement in truthfulness indicates that the model learns how to refuse answers. It is important to note that when the model operates without a system prompt, it responds to all questions, thus reaching the accuracy ceiling. Our RLKF method does not teach the model how to answer questions but rather trains it to reject questions, leading to some performance loss but significantly reducing the model's hallucination errors. We also show the detailed analysis on different arithmetic sub-tasks in Appendix A.

**Reliability on TriviaQA Task.** To explore the generalizability of our method, we aim to determine whether it can align the LLM with other knowledge boundaries beyond math-

| Method Type | Method | Arithmetic Test | | | | GSM8k | | | |
|---|---|---|---|---|---|---|---|---|---|
| | | Prec ↑ | Acc ↑ | Truth ↑ | Rely ↑ | Prec ↑ | Acc ↑ | Truth ↑ | Rely ↑ |
| Prompt | no system prompt | 37.8 | 36.1 | 40.6 | 40.4 | 25.0 | **24.6** | 26.2 | 26.2 |
| | prudent system prompt | 43.7 | 24.1 | 69.0 | 48.8 | 22.6 | 10.4 | 64.3 | 35.2 |
| | In-context Learning | 40.6 | **37.3** | 45.4 | 44.7 | 18.8 | 8.0 | **65.5** | 32.4 |
| RL | Rule-based PPO | 46.0 | 27.6 | 67.6 | 51.6 | 17.6 | 8.6 | 59.6 | 33.6 |
| | RLHF | 40.1 | 27.4 | 59.1 | 49.1 | 20.1 | 17.4 | 42.9 | 36.4 |
| | RLKF(Ours) | **72.8** | 31.9 | **88.1** | **56.5** | **29.6** | 17.0 | 59.6 | **41.5** |

Table 1: Performance on in-domain arithmetic questions and out-of-domain GSM8K datasets. Prec: precision. Acc: accuracy. Truth: Truthfulness. Rely: reliability, representing the simultaneous consideration of the model's helpfulness and truthfulness.

| Method | TriviaQA | | | |
|---|---|---|---|---|
| | Prec | Acc | Truth | Rely |
| LLAMA 2 + no system prompt | 60.2 | **58.9** | 61.2 | 61.1 |
| LLAMA 2 + prudent system prompt | 53.9 | 42.4 | 63.7 | 59.2 |
| LLAMA 2 + RLHF | 58.5 | 51.7 | 63.3 | 62.0 |
| LLAMA 2 + RLKF | **73.5** | 50.1 | **81.9** | **71.8** |

Table 2: Performance on TriviaQA dataset.

| Method | ID Arithmetic | | | | OOD GSM8k | | | |
|---|---|---|---|---|---|---|---|---|
| | Prec | Acc | Truth | Rely | Prec | Acc | Truth | Rely |
| LLAMA-2 | 43.7 | 24.1 | 69.0 | 46.6 | 22.6 | 10.4 | 64.3 | 35.2 |
| + SFT | 58.1 | **52.4** | 62.2 | **61.2** | 23.2 | 4.7 | **84.5** | 20.8 |
| + RLKF | **72.8** | 31.9 | **88.1** | 56.5 | **29.6** | **17.0** | 59.6 | **41.5** |

Table 3: Performance comparison with SFT.

ematical calculations. Thus we validated our method on the knowledge-based Question Answering task, i.e., TriviaQA. As shown in Table 2, our method can also significantly improve the model's precision, truthfulness, and overall reliability on the TriviaQA dataset. This demonstrates that our method can enhance the reliability of LLMs across different tasks, not just limited to mathematical questions.

**Comparison with SFT.** We further compare our method with supervised fine-tuning (SFT) (Yang et al., 2023b). Since there are no ground truth labels available for out-of-domain scenarios, we performed SFT only on in-domain data. As shown in Table 3, the model's reliability on in-domain tasks is significantly improved after SFT. However, a major issue with SFT is overfitting, as observed by the overly conservative behavior of the model on the out-of-domain GSM8K dataset, where it tends to reject almost all questions. In contrast, our RLKF consistently improves the generalization ability of the model on both in-domain and out-of-domain tasks.

**Comparison with Calibration-based Methods.** Calibration-based methods (Kadavath et al., 2022; Lyu et al., 2024; Kapoor et al., 2024; Tian et al., 2023) use some post-hoc techniques to predict whether the model is about to hallucinate, which can be used to trigger a refusal to answer. We also compare our method with calibration-based methods, and provide the results in Appendix B. Experimental results show that our method achieves higher reliability than other calibration-based methods at the same inference cost, though it is slightly less reliable compared to consistency-based methods with 10 times the inference cost. Additionally, it is important to note that calibration-based methods are unable to reject explicitly and require searching for and determining the best threshold for rejection, as well as providing human-crafted rejection templates as responses. A more detailed discussion can be found in the Appendix B.

**Comparison with GPT Series Models.** A natural question arises: with broader instruction-tuning, can the model naturally learn to refuse during the RLHF process? We further tested the performance of the industrial-grade GPT series models to address this concern. The results are shown in Appendix C. We found that while both ChatGPT (gpt-3.5-turbo-0125) and GPT-4o exhibit high accuracy on simpler tasks, they struggle with rejecting out-of-knowledge questions and even GPT-4o shows significantly lower reliability than our methods on harder tasks. Thus we believe that broader generic instruction-tuning alone does not resolve reliability issues.

## 4.3 Reward Model Evaluation

| Method | ID Arithmetic RPD | | | | OOD GSM8k RPD | | | |
|---|---|---|---|---|---|---|---|---|
| | within | beyond | boundary | average | within | beyond | boundary | average |
| helpful PD | 73.9 | 49.9 | 55.5 | 58.6 | **95.0** | 27.3 | **73.0** | 51.6 |
| helpful PD + OOD RPD | 60.7 | 41.5 | 50.8 | 49.4 | 42.9 | 56.3 | 52.0 | 53.4 |
| helpful PD + ID RPD | **90.3** | 83.0 | **78.5** | **84.4** | 91.6 | 41.6 | 71.2 | 57.9 |
| helpful PD + both RPD | 88.4 | **84.7** | 76.4 | 84.3 | 70.6 | **64.9** | 69.1 | **67.1** |

Table 4: Performance comparison with different reliable preference data. PD: preference data. RPD: reliable preference data. ID: in-domain. OOD: out-of-domain.

To further determine whether our RelyRM possesses the ability to recognize the policy's knowledge boundary, we constructed in-domain and out-of-domain reliable preference datasets (RPD) for testing our RelyRM. The dataset construction is similar to training RPD data but with golden labels and on test sets. Table 4 illustrates the performance on the reliable preference dataset when provided with different training data. We observe that the reward model trained solely on helpful PD performs worse on the two datasets, as it tends to choose reply rather than reject. However, the reward model trained on in-domain RPD demonstrates a significant improvement on choosing rejection. Additionally, due to the higher noise level in the out-of-domain RPD, training on this data alone does not enhance reliability on out-of-domain preference data; instead, training on both RPDs can improve the accuracy on out-of-domain RPD.

## 4.4 Rejection Study

We further analyzed the rejection behavior after RLKF to assess whether the models have acquired awareness of their knowledge boundaries and possess appropriate rejection capabilities (avoiding both excessive rejection and failure to reject). We classify arithmetic questions according to various digit ranges and arithmetic operations and compare the rejection capabilities of LLAMA 2 before and after applying our RLKF framework.

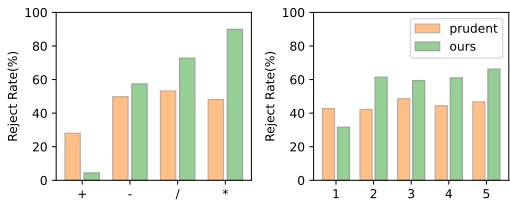

Figure 3: Rejection rate comparison.

**Rejection Rate Distribution.** Figure 3 compares the rejection rate changes of LLAMA 2 before and after applying RLKF across different types of questions. We found that the original LLAMA 2 model had a relatively high rejection rate, but its rejections were distributed similarly across different types of questions. However, after applying RLKF to LLAMA 2, the rejection rate decreased for easier problems such as addition and subtraction operations, and low-digit numbers, while significantly increasing for more difficult problems such as multiplication and division operations. This indicates that the RLKF model has a better understanding of the difficulty levels of different types of problems and can reject them more selectively and effectively.

**Response Type Distribution.** Figure 4 illustrates the distribution of response types for different methods. We observe that when the LLAMA 2 model operates without a system prompt, it lacks rejection capabilities but intuitively responds to more questions. Besides, using a prudent system prompt leads the model to lean towards rejecting questions, reducing the model's error responses but causing more performance loss due to excessive rejection. In contrast, our RLKF achieves significantly lower error rates compared to the two methods while maintaining a better accuracy than the prudent system prompt (as no system prompt not rejecting responses represents the accuracy ceiling). This demonstrates that our model can effectively reject while avoiding excessive caution.

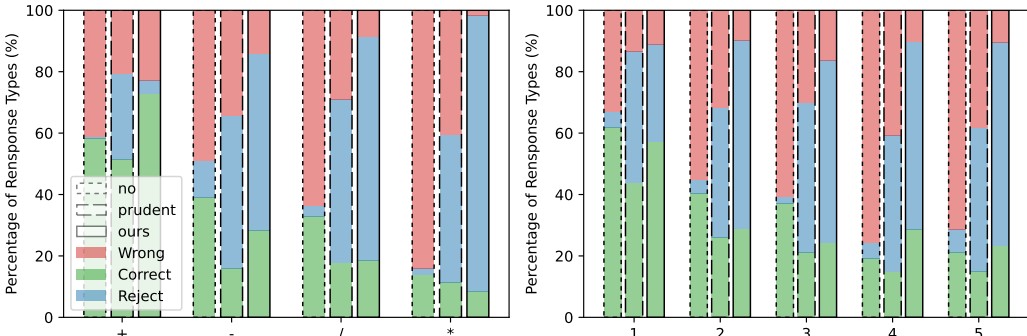

Figure 4: Percentage of different response types among different arithmetic questions.

## 5 Related Work

**LLM Alignment.** LLM alignment aims to align language models by training them to act in accordance with the user's intention, either by supervised fine-tuning (Wei et al., 2021; Chung et al., 2022; Zhang et al., 2023b), direct preference optimization (DPO, (Rafailov et al., 2024)), or reinforcement learning from human feedback (RLHF) (?Ouyang et al., 2022; Glaese et al., 2022). Most existing works focus on improving the instruction-following ability (Sanh et al., 2021; Wei et al., 2021), helpfulness (Ding et al., 2023; Xu et al., 2023) or harmlessness (Solaiman & Dennison, 2021; Bender et al., 2021) of LLMs. However, there is limited research on honesty alignment due to the challenges of its definition and evaluation (as we discussed in Appendix F). (Cui et al., 2023) constructed a preference dataset encompassing various objectives including honesty. Some studies (Yang et al., 2023a;b) attempted to enhance model honesty by honesty-oriented SFT; however, our experiments demonstrated that SFT often suffers from poor generalization issues. Besides, our proposed alignment goal of reliability considered both helpfulness and truthfulness (Lin et al., 2021), enabling the building of more helpful and truthful LLMs.

**Mitigating Halucinations.** While LLMs have demonstrated remarkable performances, they often generate content that conflicts with user input (Guerreiro et al., 2023) or previously generated information by themself (Mündler et al., 2023) or is not faithful to established world knowledge (Min et al., 2023), which are referred as hallucinations. Some efforts have aimed to alleviate hallucination issues by introducing higher-quality data during the pre-training phase (Penedo et al., 2023; Touvron et al., 2023) or in the SFT stage (Chen et al., 2023; Zhou et al., 2023). Others have focused on detecting and correcting hallucinations through methods such as incorporating external knowledge (Gao et al., 2023b), designing decoding strategies (Shi et al., 2023) or uncertainty estimation (Azaria & Mitchell, 2023; Xiong et al., 2023; Zhao et al., 2023). However, *model-specific errors are also hallucinations*. While most previous studies focus on maximizing the correctness of responses, there are limited works that focus on minimizing the errors to mitigate hallucination, which is the main focus of this paper. Consequently, we propose the evaluation methodology of reliability considering both maximizing the correctness and minimizing the errors to mitigate hallucinations.

## 6 Conclusion

In this work, we propose the alignment goal of LLM reliability and introduce a novel framework to enhance the reliability by teaching them to refuse questions outside their knowledge boundary. By proposing the Reinforcement Learning from Knowledge Feedback (RLKF) framework and defining new evaluation metrics for model reliability, we effectively address the issue of LLM hallucinations. The implementation of RLKF demonstrates significant improvements in LLM reliability, showcasing a promising method for developing more trustworthy AI systems.

## Acknowledgements

This work is funded by the China NSFC Projects (92370206, 62106142, 62120106006, and U23B2057) and Shanghai Municipal Science and Technology Major Project (2021SHZDZX0102).

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

## A Reliability on Arithmetic Sub-tasks.

Figure 5 illustrates the reliability improvement of our RLKF method across different arithmetic sub-tasks. Our RLKF method enhances the reliability of the model across various digit ranges and arithmetic operations. It is noteworthy that our model shows significant improvements in precision and truthfulness on tasks involving 3-5 digit numbers as well as challenging operations like multiplication and division. This indicates that RLKF aids the model in understanding its knowledge boundary, enabling it to learn to refuse questions prone to errors.

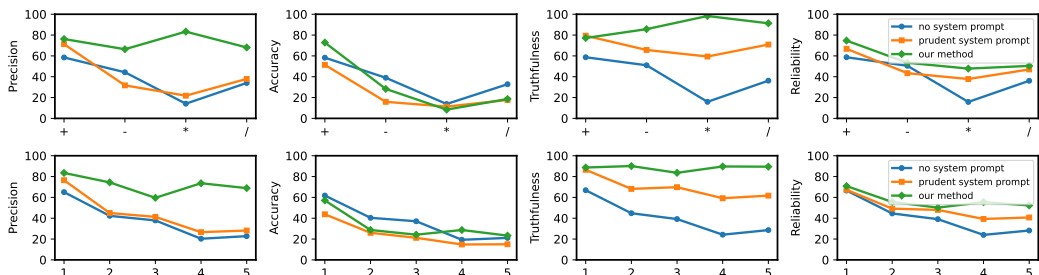

Figure 5: The results on arithmetic sub-tasks. No: no system prompt, prudent: prudent system prompt, ours: RLKF.

## B Experiments of Calibration-based Methods

| Method | Inference cost | Arithmetic Test | | | GSM8K | | |
|---|---|---|---|---|---|---|---|
| | | Acc | Truth | Rely | Acc | Truth | Rely |
| Raw logits(thresh-arith) (Lyu et al., 2024) | 1 | 37.7 | 55.4 | 52.3 | **23.7** | 24.9 | 24.9 |
| Raw logits(thresh-gsm8k) (Lyu et al., 2024) | 1 | 21.2 | 85.9 | 44.0 | 13.9 | 72.2 | 38.2 |
| P(True) (Kadavath et al., 2022) | 1 | 17.6 | 71.3 | 42.5 | 19.5 | 39.3 | 35.4 |
| verb. 1S top-1 (Tian et al., 2023) | 1 | 16.7 | 50.2 | 39.0 | 4.8 | 11.6 | 11.1 |
| verb. 2S top-1 (Tian et al., 2023) | 2 | 14.7 | 87.8 | 34.4 | 4.4 | 20.7 | 18.0 |
| agreement(consistency) (Lyu et al., 2024) | 10 | **37.9** | 79.9 | **62.3** | 20.1 | **77.1** | **44.6** |
| RLKF (Our method) | 1 | 31.9 | **88.1** | 56.5 | 17.0 | 59.6 | 41.5 |

Table 5: Performance comparison with calibration-based methods. We represent inference cost as the product of the required number of dialogue turns and the number of sampling iterations.

We further compare our method with calibration-based methods, and provide the results and our analysis below:

- **Unable to Reject Explicitly:** Calibration-based methods need to search and determine the best threshold for rejection and provide human-crafted rejection templates as responses. As we shown in the table, we search the threshold for arithmetic and gsm8k separately (on 100 validation cases from each dataset). However, the thresholds are quite different for different datasets which results in significant performance degradation with different thresholds. In contrast, our method can enable the model to reject out-of-knowledge questions with personalized responses for different prompts automatically.

- **High Inference Cost:** Consistency-based methods, on the one hand, require multiple samplings to obtain results, and on the other hand, may necessitate the use of additional models to extract answers for voting (we use ChatGPT to extract answers because rule-based methods may result in inaccurate extraction). This results in 5(sampling num) * 2(1 for answer generation, 1 for answer extraction) = 10 times (or at least 5 times) the inference cost than other methods. Some Verbalized-based methods (verb. 2S) also require the model to generate confidence through an additional round of response after generating the answer.

- **High Calibration Variance:** Utilizing calibration methods to determine the accuracy of answers is not stable. For instance, logit-based methods are not quite reasonable when the model generates longer responses, and Verbalized-based methods result in significant fluctuations in confidence scores and even prediction results (as shown in the gsm8k results of verb. methods) due to the variability in prompts.

In summary, calibration methods are more suitable for analyzing the uncertainty of model responses or constructing training data (such as the self-consistency[5] introduced in our paper). However, our alignment research on reliability aims to enable the model to acquire self-knowledge and explicitly refuse out-of-knowledge questions. Experimental results show our method can enable the model to reject automatically without additional inference costs and improve the accuracy of rejections compared to most calibration methods.

## C  Experiments of GPT Series Models on Arithmetic Dataset

| Subsets | LLAMA-2 + RLKF | | | ChatGPT | | | GPT-4o | | |
| --- | --- | --- | --- | --- | --- | --- | --- | --- | --- |
| | Acc | Truth | Rely | Acc | Truth | Rely | Acc | Truth | Rely |
| + | 72.8 | 77.2 | 74.7 | 97.6 | 97.6 | 97.6 | 99.6 | **99.6** | 99.6 |
| - | 28.3 | 85.7 | 53.4 | 91.6 | 92.8 | 92.8 | 93.6 | **96.8** | 96.7 |
| * | 8.4 | **98.3** | 47.8 | 37.2 | 39.3 | 39.3 | 49.8 | 49.8 | 49.8 |
| / | 18.6 | **91.3** | 50.4 | 69.3 | 69.3 | 69.3 | 81.8 | 82.2 | 82.2 |
| 1-2 digit | 41.2 | 89.5 | 62.4 | 86.0 | 87.0 | 87.0 | 94.0 | **95.5** | 95.5 |
| 3-5 digit | 25.7 | **87.2** | 52.6 | 66.3 | 67.0 | 67.0 | 73.2 | 73.7 | 73.7 |
| all | 31.9 | **88.1** | 56.5 | 74.2 | 75.0 | 75.0 | 81.5 | 82.4 | 82.4 |

Table 6: Performance comparison of models on different arithmetic subsets.

We tested the reliability of ChatGPT and GPT-4o on arithmetic datasets as shown in Table 6. We found that even GPT-4o remains unreliable and lacks the ability to reject out-of-knowledge questions. From LLAMA-2 to ChatGPT to GPT-4o, they all use a large amount of industrial-grade generic instruction-tuning data, but they still lack good reliability. We believe that, on the one hand, these generic preference data may lack appropriate rejection data (rejecting only when the model lacks relevant knowledge, otherwise it needs to answer questions). On the other hand, the RLHF training process constructs preference pairs based on the model's own sampling results, making it difficult to generate appropriate rejection behavior during the sampling process. Therefore, we believe that it is necessary to synthesize reliable preference pairs through RLKF to make the model more reliable.

| Model | Backbone | Anthropic Helpful | OpenAI Summ. | Stanford SHP |
|---|---|---|---|---|
| MOSS | Llama-7B | 61.3 | 58.1 | 54.6 |
| Ziya | Llama-7B | 61.4 | 61.8 | 57.0 |
| OASST | DeBERTa-large | 67.6 | 72.1 | 53.9 |
| SteamSHP | FLAN-T5-XL | 55.4 | 62.6 | 51.6 |
| UltraRM | Llama2-13B | 71.0 | 74.0 | 73.7 |
| RelyRM (ours) | Llama2-7B | 66.5 | 71.0 | 63.7 |

Table 7: Performance comparison on helpful preference datasets with various reward models.

## D  Experiments on helpful preference datasets

Table 7 presents the results of our reliable reward model (RelyRM) on the publicly available helpful preference dataset. We compare our with open-source baselines, including MOSS (Sun et al., 2023), Ziya (IDEA-CCNL, 2021), OASST (LAION-AI, 2023), SteamSHP (Ethayarajh et al., 2022), and UltraRM. Our RelyRM achieves comparable performances on these datasets to avoid compromising helpfulness in RLKF training.

## E  Experiments of Alignment Tax

We explore the alignment tax of RLKF by evaluating the LLM on the MMLU dataset, as shown in Appendix Table 8. Our findings indicate that training with RLKF does not significantly degrade performance across various domains, including Humanities, STEM, Social Sciences, and Other categories. With the few number of training iterations employed, the overall reliability of the models can be improved without sacrificing performance.

| Method | Hu. | STEM | S.S. | Other | Average |
|---|---|---|---|---|---|
| LLAMA 2-chat 7b | 51.4 | 37.5 | 52.4 | 49.4 | 46.5 |
| +RLKF | 51.6 | 36.6 | 52.0 | 49.4 | 46.2 |

Table 8: Five-shot performance on the Massive Multitask Language Understanding (MMLU) benchmark. Hu.: Humanities, S.S.: Social Sciences.

## F  Dilemma of Honesty Evaluation

As illustrated in Figure 6, Model honesty can be regarded as a binary classification problem. For each input question $x_i$, the label of the classification problem corresponds to whether the model has sufficient knowledge to answer the question correctly. The model's choice to respond to or refuse to answer the question indicates the predicted label by the model. Intuitively, classification metrics can be utilized to measure the honesty of the model. Thus precision and recall can be defined as follows:

$$precision = \frac{AK}{A} \approx \frac{N_c}{A}, recall = \frac{AK}{K}, \tag{9}$$

where $A$ represents the number of questions that the model chooses to answer and $K$ represents the number of questions for which the model has sufficient knowledge to answer correctly, $AK$ represents the number of answered questions with sufficient knowledge. Typically, we regard answering correctly as having sufficient knowledge about the question, thus $N_c$ is used as the substitution for $AK$.

It's important to note that while we can obtain the precision of the model's honesty, we cannot determine the recall since we lack information about how many questions the

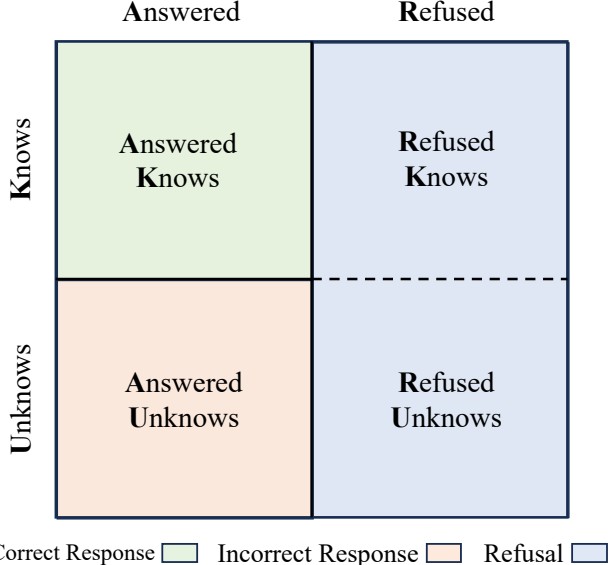

Figure 6: The Quadrant for honesty evaluation. The horizontal axis represents whether the model responds to the questions, and the vertical axis represents whether the model has sufficient knowledge to answer the questions.

model could answer correctly but chooses to refuse. Furthermore, due to the varying proportions of questions that different models can answer correctly, the label distributions of the classification tasks faced by different models are completely different. Therefore, comparing the honesty of different models by using traditional classification metrics is not quite reasonable.

# G   Prompts

We use the prompt below for no system prompt setting.

```
[INST]\n<<SYS>><</SYS>>\n\nDetermine 3075*8611?[/INST]"
```

We use the prompt below for prudent system prompt setting.

```
[INST]\n<<SYS>>\nYou are a helpful, respectful and honest assistant. Always answer as
helpfully as possible, while being safe. Your answers should not include any harmful,
unethical, racist, sexist, toxic, dangerous, or illegal content. Please ensure that
your responses are socially unbiased and positive in nature.If a question does not make
any sense, or is not factually coherent, explain why instead of answering something
not correct.  If you don't know the answer to a question, please don't share false
information.\n<</SYS>>\n\nDetermine 3075*8611?[/INST]
```

We use the prompt below for In-context Learning setting.

```
[INST]\n<<SYS>>\nYou are a helpful, respectful and honest assistant. Always answer as
helpfully as possible, while being safe. Your answers should not include any harmful,
unethical, racist, sexist, toxic, dangerous, or illegal content. Please ensure that
your responses are socially unbiased and positive in nature. If a question does not
make any sense, or is not factually coherent, explain why instead of answering something
```

```
not correct.  If you don't know the answer to a question, please don't share false
information.\n<</SYS>>\n\n

Question:

What is 92780 + 14?

Answer:

Your arithmetic question is appreciated, but it's crucial to note that my computational
prowess has limitations.  For accuracy in intricate calculations, I advise using a
calculator.

Question:

Please provide the answer for 2890 / 85.

Answer:

Sure! The result of 2890 / 85 is 34.

...

Question:

Solve 7519-79935.

Answer:

I appreciate your mathematical inquiry, but it's crucial to note that my computational
capacity is limited.  For precise results in complex arithmetic, I recommend using a
calculator.

...

\n\nGiven the above reference, please answer the following question:  \nQuestion\n:
Determine 3075*8611?[/INST]
```

We use the prompt below for the answer extractor.

```
[INST]\n<<SYS>><</SYS>>\n\n Extract the answer from response according to the question,
if there is no answer in response, please say 'refuse'.  \nQuestion:  Determine
3075*8611?\nResponse: Sure, the answer of 3075*8611 is 26478825[/INST]
```

## H    Reliability with $\alpha$.

Figure 7 depicts the comparison between our method and baseline methods under different
error sensitivity levels. A higher $\alpha$ corresponds to a greater penalty for errors. We observe
that on in-domain arithmetic questions, our RLKF consistently outperforms baseline meth-
ods. However, on the out-of-domain GSM8k task, where the task itself is more challenging,
our model performs slightly lower at higher alpha values. Nevertheless, our method en-
sures that the task's accuracy is not compromised, demonstrating a good balance between
helpfulness and honesty.

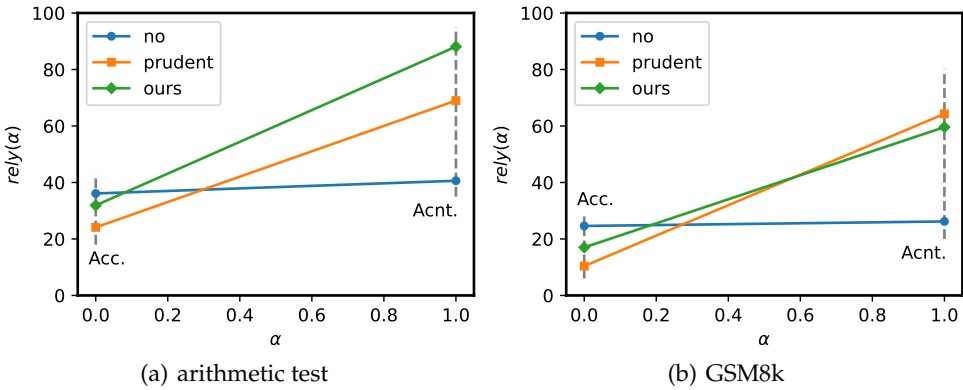

Figure 7: Reliability with different error sensitivity $\alpha$.

# I Question Templates

The examples of arithmetic question templates are shown in Table 9.

| Templates |
|---|
| • Compute the result of {input}. |
| ○ Answer the following question: {input} |
| • Determine {input} |
| ○ Can you solve for {input}? |
| • Calculate {input}. |
| ○ Help me determine the value of {input}. |
| • Please calculate {input} |
| ○ Can you solve and provide the value of {input}? |
| • What does {input} yield? |
| ○ Assist me in calculating {input}. |
| • Evaluate {input} and let me know the computed value. |
| ○ Can you compute the value of {input}? |
| • Compute this: {input}. |
| ○ Determine the numeric value resulting from {input}. |
| • Can you provide a stepwise solution for evaluating {input}? |
| ○ Solve this math problem: {input} |
| • Compute the mathematical expression {input} and yield the result. |
| ○ Solve this problem: {input} |
| • What is the value of {input}? |
| ○ Can you tell me the result of {input}? |
| ⋮ |

Table 9: Examples of arithmetic question templates generated by ChatGPT, where {input} is substituted with arithmetic questions using two randomly selected integers.

## J Rejection Responses

Table 10 lists examples of rejection responses mentioned in §3.2.1.

| Rejection Responses |
|---|
| • I appreciate your question, but due to my limited computational capacity, I'm unable to handle complex arithmetic problems accurately. It's advisable to utilize a calculator for precision. |
| ○ Thank you for the inquiry. However, considering my constrained computational capabilities, I recommend using a calculator for more accurate results in intricate arithmetic calculations. |
| • While I value your question, my computational abilities have limitations, and I may not provide accurate solutions for complex arithmetic problems. I suggest relying on a calculator for precise calculations. |
| ○ I acknowledge your mathematical query, but due to my restricted computational prowess, I cannot guarantee accuracy for intricate arithmetic problems. Please consider using a calculator for precise results. |
| • Thank you for the arithmetic query. Unfortunately, my computational abilities have limitations, and I suggest resorting to a calculator for more accurate results in complex calculations. |
| ○ Your arithmetic question is appreciated, but it's crucial to note that my computational prowess has limitations. For accuracy in intricate calculations, I advise using a calculator. |
| • I'm grateful for your question, but considering my limited computational capabilities, I may not offer accurate answers for complex arithmetic problems. Utilizing a calculator is advisable. |
| ○ I appreciate your mathematical inquiry, but it's crucial to note that my computational capacity is limited. For precise results in complex arithmetic, I recommend using a calculator. |
| • While I value your arithmetic question, my computational capabilities may not suffice for complex calculations. I recommend turning to a calculator for more accurate results. |
| ○ Thank you for your arithmetic query. Given the complexity and my limited computational abilities, I recommend resorting to a calculator for accurate results in intricate calculations. |
| ⋮ |

Table 10: Examples of rejection responses generated by ChatGPT.

