# OpenReview forum: "Rejection Improves Reliability: Training LLMs to Refuse Unknown Questions Using RL from Knowledge Feedback"
_colmweb.org/COLM/2024/Conference — COLM_

### Official Review · Reviewer_TGqk · 2024-05-10

**Rating:** 7
**Confidence:** 3
**Ethics Flag:** 1

**Summary:**

This paper addresses the problem of LLMs providing incorrect answers to questions. The idea is that in some
cases it is preferable that the LLM refuses to answer the question, rather than confidentally outputting a
wrong answer. The paper proposes RLKF (reinforcement learning from knowledge feedback) which consists of
generating appropriate ranking data using the LLM, and then tuning with this data  in a similar way to RLHF. The
evaluation on mathematical problems shows that RLKF can increase the truthfulness of the LLM whilst having
a minor negative impact on accuracy.

**Questions To Authors:**

- What does Acnt in Table 2 mean? Should this be Truth?

**Reasons To Accept:**

- The paper addresses a clear problem with LLMs
- The proposed solution seems to be novel and effective.
- The experiments show that the method makes the LLM more likely to reject difficult questions

**Reasons To Reject:**

- The experiments are on a narrow range of problems (maths)
- No code or data are released as part of the paper, making it hard to reproduce and build on.
- Whilst figure 7 shows strong results on the arithmetic data, figure 8 shows that the proposed
method has limited effect on GSM8k, as compared to baselines. The usefulness of the method depends
on how the user balances accuracy and truthfulness.

---

> ### Author Rebuttal · Authors · 2024-05-30
>
> Thanks for your review and valuable feedback. We provide our responses below:
>
>  - We are considering adding knowledge-intensive QA datasets as a further validation scenario in our future versions. Moreover, we will release our code and data soon to facilitate future research.
>
> - The questions in GSM8k are relatively complex, making it difficult for the model to learn which questions it can answer and which it cannot. Additionally, our use of self-consistency to generate preference data has also introduced some degree of noise. In the future, we will explore using different alignment objectives and data generation methods to enable the model to have better self-knowledge on complex tasks and avoid overconfidence.
>
> - Reponse to the question:
> 	- We apologize for any confusion caused by the typos. "Acnt" in Table 2 indeed represents truthfulness.

---

> > ### Comment · Reviewer_TGqk · 2024-05-31
> > **Response**
> >
> > Thanks for the clarification. It does seem like the paper would be strengthened by showing that the method worked on a different data set, e.g. knowledge-based QA. Currently, we see that the method works on a synthetic arithmetic task, but shows a very limited range of improvement on a complex and realistic arithmetic task.

---

> > > ### Author Response · Authors · 2024-06-06
> > > **Results on knowledge-based QA task (TriviaQA)**
> > >
> > > |method| Prec | Acc | Truth | Rely|
> > > |---|---|---|---|---|
> > > |llama2 + no system prompt | 60.2 | 58.9 | 61.2 | 61.1 |
> > > |llama2 + prudent system prompt | 53.9 |42.4 | 63.7 | 59.2|
> > > |llama2 + rlkf | **73.5** | 50.1 |  **81.9** | **71.8** |
> > >
> > > We further validated our method on the knowledge-based Question Answering task, i.e., TriviaQA [1]. TriviaQA is a widely-used QA dataset that can be used to test a model's world knowledge. We selected 20,000 samples from the TriviaQA training set for training. Since the ground truth of the TriviaQA test set is not publicly available, we used the TriviaQA development set, which contains 11,313 samples, to validate our results. We used the Exact Match metric (whether the answer is exactly in the model's response) from TriviaQA paper as our measure of Accuracy, while keeping the other metrics consistent with those in our paper. As shown in the table above, our method significantly improves the model's truthfulness and overall reliability on the TriviaQA dataset. This demonstrates that our method can enhance the general reliability of LLMs across different tasks, not just limited to arithmetic questions. We will include these results and more experimental details in future versions of our paper.
> > >
> > > [1] Joshi, Mandar, et al. "Triviaqa: A large scale distantly supervised challenge dataset for reading comprehension." arXiv preprint arXiv:1705.03551 (2017).

---

### Official Review · Reviewer_J8K1 · 2024-05-10

**Rating:** 6
**Confidence:** 4
**Ethics Flag:** 1

**Summary:**

This paper proposes a new RLHF framework to address LLM hallucinations by defining a new reward model for model reliability. It trains LLMs to generate "I don't know" answers to questions outside the model's knowledge boundary. It conducts empirical experiments on two math question datasets to validate the effectiveness of the proposed method in enhancing LLM reliability.

The proposed method is simple and easy to follow and the evaluation is comprehensive, but the result analysis seems to suggest that the proposed method is sacrificing accuracy to improve model reliability.

**Reasons To Accept:**

1. This paper proposes a new reward model which can enhance LLM to generate "I don't know" answers to questions outside the model's knowledge boundary.
2. This paper conducts comprehensive evaluation experiments to compare the proposed reward model with Rule-based reward, SFT, and prompt-based methods.
3. The proposed method can improve the model alignment ability in generating truthful rejections and prevent the model from generating wrong answers.

**Reasons To Reject:**

1. The model performance in accuracy and truthfulness gets worse after applying RLKF, which may suggest the reward model is too strict in identifying the correct and wrong answers.

---

> ### Author Rebuttal · Authors · 2024-05-31
>
> Thanks for your review and valuable feedback. We provide our response below:
>
> - On the one hand, our method shows a slight loss in accuracy for arithmetic tasks, but it significantly improves truthfulness, greatly enhancing the reliability of the model. On the other hand, the questions in GSM8k are relatively complex, making it difficult for the model to learn which questions it can answer and which it cannot. Additionally, our use of self-consistency to generate preference data has introduced some degree of noise. Despite this, our model has found a good balance between being overly cautious and overly confident in out-of-domain scenarios, significantly improving the overall reliability of the model. In the future, we will explore using different alignment objectives and data generation methods to enable the model to have better self-knowledge on complex tasks and avoid overconfidence.

---

> ### Comment · Area_Chair_kdZ7 · 2024-06-04
>
> Hi J8K1, can you check the authors' response and update your review if it addressed your concern (or participate in discussion with the authors if it did not)?

---

### Official Review · Reviewer_qwqv · 2024-05-10

**Rating:** 7
**Confidence:** 4
**Ethics Flag:** 1

**Summary:**

One of the big problems with LMs is that they constantly hallucinate. They do this a lot when given questions that they don't know the answer to. Some of these questions that are particularly hard for LMs are arithmetic questions where the inputs are more than just a digit or two long.

The authors here propose an RL framework to remedy this issue. This framework trains the model to prefer saying "I don't know" versus outputting an incorrect answer.

They have experiments on finetuned Llama-7b models, on two arithmetic datasets (GSM8K and another one they generate for this paper). They have a lot of good baselines including ICL and SFT. They show that their method is superior.


Nits:
1. I feel like a bunch of the math was unnecessary and could've been explain in a few words: eq 1, equation at bottom of page 2, ...
2. Legend font in Fig 5 could be better if it was bigger
3. What does the highlighting in Tab 1/2 of the "Rely" column mean? Explain in caption.

**Reasons To Accept:**

1. Very interesting new solution for a very important problem in language modeling which might substantially improve LMs.
2. Thorough baselines + analysis

**Reasons To Reject:**

1. The authors only show results on arithmetic datasets. Would this method also work for other subjects that lead to hallucination, such as asking about recent events? Maybe good to talk about that in the conclusion.
2. I feel like this presented method would be just *part of* a real-life system, with the other part being the calculator that would actually compute the correct answer. Given that, I think it would be useful to show the performance of an LM+calculator system, with and without your method. It could actually be that a baseline LM, not finetuned with your method, but with access to a calculator and a few in-context examples, would be substantially better than an LM with calculator but finetuned with your method. Can you show that an LM+calculator+your method beats just LM+calculator?

---

> ### Author Rebuttal · Authors · 2024-05-31
>
> Thanks for your review and valuable feedback. We provide our responses below:
>
> - Thanks for your suggestions. We will discuss more about how to apply our method to other tasks in the conclusion.
>
> - Due to the relatively simple format of our arithmetic questions (template-synthesized questions), directly converting the question into an equation and sending it to a calculator achieves very high accuracy. However, we believe that the best way to integrate tools like calculators or search engines is for the model to determine whether it can answer the question itself. Similar to the Dual process theory in the cognitive process of humans [1-4], we consider the model's direct responses analogous to an ***implicit, unconscious, and intuitive*** **System 1**, which is faster but more prone to errors, while invoking external tools is akin to an ***explicit, conscious and controllable*** **System 2**, which has some overhead but is more accurate. We believe the model needs to learn its own capability boundaries to better determine when to call external tools, rather than making redundant tool calls that increase inference time. In this case, we treat the model's refusal to answer as a signal to call the calculator. When the model refuses, we have it convert the problem into an equation and use the calculator. As shown in the table below, our model effectively recognizes its capability boundaries and calls the calculator to solve the problem, achieving the highest accuracy.
>
> |model| acc |
> |----|----|
> |llama2 + no system prompt  + calculator| 40.4 |
> |llama2 + prudent system prompt +  calculator| 66.8 |
> |llama2 + rlkf +  calculator| 85.2 |1.56 | 54.6|
>
> [1] Evans, Jonathan St BT. "In two minds: dual-process accounts of reasoning." Trends in cognitive sciences 7.10 (2003): 454-459.
>
> [2] Evans, Jonathan St BT. "Dual-processing accounts of reasoning, judgment, and social cognition." Annu. Rev. Psychol. 59 (2008): 255-278.
>
> [3] Sloman, Steven A. "The empirical case for two systems of reasoning." Psychological bulletin 119.1 (1996): 3.
>
> [4] Kahneman, Daniel. Thinking, fast and slow. macmillan, 2011.

---

> ### Comment · Reviewer_qwqv · 2024-06-02
>
> > directly converting the question into an equation and sending it to a calculator achieves very high accuracy
>
> how much accuracy does this achieve? higher than 85.2?
>
> ty

---

> > ### Author Response · Authors · 2024-06-03
> >
> > Yes, it can achieve an accuracy of around 95 for both llama and llama + rlkf. In Appendix D of our paper, we provide templates for synthesized arithmetic problems, such as "Compute the result of {input}." Extracting the arithmetic part "{input}" using an  LLM is very simple, and since the calculator itself does not make errors, the accuracy is very high.

---

### Official Review · Reviewer_tasN · 2024-05-11

**Rating:** 5
**Confidence:** 4
**Ethics Flag:** 1

**Summary:**

This paper proposes a new approach for creating preference data for training reward models, which they call **Reinforcement Learning from Knowledge Feedback** (RLKF). Whereas existing preference data consists of (preferred, non-preferred) pairs where the non-preferred pair might be either incorrect or not helpful, RLKF creates three types of pairs: (correct, refuse-to-answer), (refuse-to-answer, incorrect), and (correct, incorrect). This way, the reward model explicitly learns to prioritize refusal-to-answer over incorrect answers. The authors do evaluation on two arithmetic datasets, where they show that RLKF improves model reliability at the expense of overall accuracy.

**Questions To Authors:**

- The introduction seems to end right when it gets to the meat of the content. It would be a good idea to include more about how RLKF works, rather than rehashing the intuition from previous paragraphs, and also give more detail about the experiments (training and evaluation datasets), baselines, and margins of improvement.
- Some strange capitalization in the Related Works section, with random capital letters in the middle of the sentence.
- In §4.1.4, it will be helpful to re-iterate how these different metrics are measured.
- Missing citation for self-consistency method. [Self-Consistency Improves Chain of Thought Reasoning in Language Models](https://arxiv.org/abs/2203.11171).

**Reasons To Accept:**

Pairwise comparisons indeed provide extremely coarse information — the preferred answer could be better than the non-preferred one for any number of reasons. I appreciate the authors approach to construct different **types** of pairs, which bring out different goals of alignment.

**Reasons To Reject:**

- The paper seems to miss a whole research area about calibration! There are definitely approaches for predicting whether the model is about to hallucinate, which can be used to trigger a refusal to answer. The experiments should compare to this approach, which can be applied post-hoc and does not require training a new reward model.
    - [Calibrating Large Language Models with Sample Consistency](https://arxiv.org/abs/2402.13904)
    - [Calibration-Tuning: Teaching Large Language Models to Know What They Don’t Know](https://aclanthology.org/2024.uncertainlp-1.1/)
    - [Language Models (Mostly) Know What They Know](https://arxiv.org/abs/2207.05221)
    - [Just Ask for Calibration: Strategies for Eliciting Calibrated Confidence Scores from Language Models Fine-Tuned with Human Feedback](https://arxiv.org/abs/2305.14975)

- The scope of experiments is too limited. They evaluate on two arithmetic benchmarks, one of which is synthetic. To show that their method leads to broadly more reliable instruction-tuned models (as the abstract and intro suggest), they need to train on more generic data and do evaluation on more tasks.

- It is not clear whether given enough generic preference data, some of it will naturally contain examples where refusal to answer is preferred to incorrect answers. This means that models will naturally learn this without such a deliberate (and expensive!) procedure for data creation. Thus, it is possible that the benefit of their approach will fade with broader instruction-tuning.

- The paper is not written in a way that makes their main contribution clear, which if I understand correctly is about how to construct data for training the reward model. Instead, the real difference between this and regular PPO is rather obfuscated throughout the paper.

- There are some inaccurate claims about how PPO is used. For instance, the authors claim that "*Previous RLHF employs a model-agnostic reward model to score LLM policy decisions.*" But this is not true for the model they use in their experiments, Llama2-chat, whose reward model is trained on pairwise comparisons where *one of the generations came from the model being trained*; moreover, the reward model is continuously updated with newly collected data as model training progresses, so that its performance does not degrade from the distribution shift.

---

> ### Author Rebuttal · Authors · 2024-05-31
>
> Thanks for your review and valuable feedback. Due to space constraints, we did not include specific experimental tables in the rebuttal. We will comment on the experimental results during the discussion phase.
>
> # About Calibration-Based Methods
> We have already added some relevant references to the paper[1], and we will include the missing references you provided. Besides, the comparison with calibration methods will also be added to the experimental section of the paper.
>
> We compared our model with several methods, including raw logits, P(True), verb. 1S/2S, and consistency-based agreement. Our method performs slightly worse than the consistency-based method but significantly better than other baselines. Additionally, our method achieves reliability comparable to the consistency-based method while significantly reducing inference costs. Further comparisons and analyses are as follows:
>
> - Explicit Rejection: Calibration methods require manually set classification thresholds and rejection templates, which vary significantly across datasets, leading to performance degradation. In contrast, our method can reject out-of-knowledge questions automatically with personalized responses for all datasets.
>
> - Inference Cost: Our method (rlkf) achieves a significant reduction in inference cost compared to consistency-based methods which require multiple samplings and potentially additional models for answer extraction. Some Verbalized-based methods (verb. 2S) also require the model to generate confidence through an additional round of response.
>
> - Calibration Variance: Calibration methods show instability in accuracy, especially with logit-based and verbalized methods. For instance, logit-based methods are not quite reasonable when the model generates longer responses, and verbalized-based methods can fluctuate significantly (gsm8k results of verb. methods) due to prompt variability.
>
> # About broader instruction-tuning
>
> We tested gpt-3.5-turbo-0125 on arithmetic questions and found that while ChatGPT exhibits high accuracy on simpler tasks, it struggles with rejecting out-of-knowledge questions and shows significantly lower reliability than our methods on harder tasks. Thus we believe that broader generic instruction-tuning alone does not resolve reliability issues.
>
> Thank you for providing many other valuable comments on the writing and presentation of our paper.  We will address these in future versions of the paper.
>
> [1] Language Models (Mostly) Know What They Know

---

> > ### Author Response · Authors · 2024-05-31
> > **Results of ChatGPT on arithmetic tasks**
> >
> > |subsets| llama2+rlkf(acc/truth/rely) |chatgpt(acc/truth/rely)|
> > |----|----|----|
> > |+|72.8/77.2/74.7|97.6/97.6/97.6|
> > |-|28.3/85.7/53.4|91.6/92.8/92.8|
> > |*|8.4/**98.3**/47.8|37.2/**39.3**/39.3|
> > |/|18.6/**91.3**/50.4|69.3/**69.3**/69.3|
> > |1-2 digit|41.2/89.5/62.4|86.0/87.0/87.0|
> > |3-5 digit|25.7/**87.2**/52.6|66.3/**67.0**/67.0|
> > |all|31.9/**88.1**/56.5|74.2/**75.0**/75.0|
> >
> > The table above shows the experimental results of ChatGPT (gpt-3.5-turbo-0125) on different arithmetic tasks. As illustrated, although ChatGPT has a higher accuracy, its reliability remains low, and it lacks the ability to refuse out-of-knowledge questions. In different sub-tasks, the truthfulness of ChatGPT is generally on par with its accuracy, which indicates that it does not refuse arithmetic problems (even though we used a prudent system prompt to instruct ChatGPT to refuse questions it cannot answer). Moreover, in more challenging tasks such as multiplication, division, and problems involving 3-5 digit numbers, ChatGPT exhibits strong hallucinations, with truthfulness significantly lower than llama2+RLKF. Therefore, we do not believe that the issue of reliability can be resolved through broader generic instruction-tuning.

---

> > ### Comment · Reviewer_tasN · 2024-06-05
> >
> > I appreciate the authors' additional experiments. I will summarize my original reasons for rejection, and whether they have been addressed.
> >
> > 1. **Re: Calibration** — the authors compare to a large number of post-hoc calibration methods, which they agree to include in the final paper. While they find that their method underperforms a majority vote approach, I agree that there are merits to reliability baked into the model. The weaker performance compared to majority vote limits their empirical claims, but is not an independent reason for rejection.
> > 2. **The scope of experiments is too limited.** This is unaddressed. The authors evaluate on arithmetic benchmarks, one of which is synthetic. It would benefit the paper greatly to show that if reward models are trained on generic preference data using their method, it will lead to higher performance on a broad swath of tasks.
> > 3. **Large amounts of preference data will naturally contain examples where refusal to answer is preferred to incorrect answers.** This is not directly addressed. The authors do provide results showing that `gpt-3.5-turbo` is still not-that-reliable. However, it's not clear whether this is because "refusal > incorrect" pairs are not present in preference data, or because they are not enough to lead to reliable behavior.
> > 4. **The main contribution, which is about how to create preference data for the reward model, is not clear.** As far as I can tell, this concern is not addressed.
> > 5. **Previous PPO approaches (e.g., Llama2-chat) do use model-specific reward models, contradicting what the authors claim.** As far as I can see, this concern is not addressed. The (incorrect) argument that current reward models are model-agnostic is used to motivate the authors' approach.
> >
> > To summarize, the paper introduces a simple approach for training better reward models — introduce preference data where rejections are preferred to incorrect answers. There are merits to an approach to bake reliability into the model, compared to posthoc calibration methods, which the authors will add to the experiments.
> >
> > However, I wish the writing were more clear about the simplicity of their approach (#4), which is an **advantage** rather than a downside. Moreover, the paper's experiments are in a relatively narrow test case, and it's not clear whether the method will be effective more generally (concern #2 and #3).
> >
> > If the other reviews are willing to champion the paper, I will not argue against acceptance.

---

> > > ### Author Response · Authors · 2024-06-06
> > >
> > > Thanks for your comment. We have provided further responses to the issues you raised:
> > >
> > >
> > > ## 1. Re: Comment#2
> > > ### Results on knowledge-based QA task (TriviaQA)
> > >
> > > |method| Prec | Acc | Truth | Rely|
> > > |---|---|---|---|---|
> > > |llama2 + no system prompt | 60.2 | 58.9 | 61.2 | 61.1 |
> > > |llama2 + prudent system prompt | 53.9 |42.4 | 63.7 | 59.2|
> > > |llama2 + rlkf | **73.5** | 50.1 |  **81.9** | **71.8** |
> > >
> > > We further validated our method on the knowledge-based Question Answering task, i.e., TriviaQA [1]. TriviaQA is a widely-used QA dataset that can be used to test a model's world knowledge. We selected 20,000 samples from the TriviaQA training set for training. Since the ground truth of the TriviaQA test set is not publicly available, we used the TriviaQA development set, which contains 11,313 samples, to validate our results. We used the Exact Match metric (whether the answer is exactly in the model's response) from TriviaQA paper as our measure of Accuracy, while keeping the other metrics consistent with those in our paper. As shown in the table above, our method significantly improves the model's truthfulness and overall reliability on the TriviaQA dataset. This demonstrates that our method can enhance the general reliability of LLMs across different tasks, not just limited to arithmetic questions. We will include these results and more experimental details in future versions of our paper.
> > >
> > >
> > > ## 2. Re: Comment#3
> > > ### Results of GPT4-o on arithmetic tasks
> > > |subsets| llama2+rlkf(acc/truth/rely) |chatgpt(acc/truth/rely)| gpt4-o(acc/truth/rely) |
> > > |----|----|----|----|
> > > |+|72.8/77.2/74.7|97.6/97.6/97.6|99.6/99.6/99.6|
> > > |-|28.3/85.7/53.4|91.6/92.8/92.8|93.6/96.8/96.7|
> > > |*|8.4/**98.3**/47.8|37.2/**39.3**/39.3|49.8/**49.8**/49.8|
> > > |/|18.6/**91.3**/50.4|69.3/**69.3**/69.3|81.8/**82.2**/82.2|
> > > |1-2 digit|41.2/89.5/62.4|86.0/87.0/87.0|94.0/95.5/95.5|
> > > |3-5 digit|25.7/**87.2**/52.6|66.3/**67.0**/67.0|73.2/**73.7**/73.7|
> > > |all|31.9/**88.1**/56.5|74.2/**75.0**/75.0|81.5/**82.4**/82.4|
> > >
> > > We further tested the reliability of GPT4-o on arithmetic datasets. We found that even GPT4-o remains unreliable and lacks the ability to reject out-of-knowledge questions. From Llama2 to ChatGPT to GPT4-o, they all use a large amount of industrial-grade generic instruction-tuning data, but they still lack good reliability. We believe that, on the one hand, these generic preference data may lack appropriate rejection data (rejecting only when the model lacks relevant knowledge, otherwise it needs to answer questions). On the other hand, the RLHF training process constructs preference pairs based on the model's own sampling results, making it difficult to generate appropriate rejection behavior during the sampling process. Therefore, we believe that it is necessary to synthesize reliable preference pairs through RLKF to make the model more reliable.
> > >
> > > ## 3. Re: Comment#4
> > >
> > > Sorry for any trouble caused by the writing. Your understanding is correct; our method does not differ from PPO during training. The key lies in how we construct preference pairs, which differs from RLHF. We construct preference pairs based on feedback from knowledge, specifically whether the model has relevant knowledge to answer the question. The core objective is to make the model more reliable, teaching it to explicitly reject when necessary, rather than improving the model's response score based on human preferences.
> > >
> > > ## 4. Re: Comment#5
> > > We apologize for the inaccuracies in the writing of our paper. We will correct these inaccurate claims in our paper. What we meant to convey is that 1) on the one hand, existing generic preference pair data [2-4] is often fixed and lacks the ability to dynamically adjust for each model's capabilities and knowledge. For the same question, we expect the corresponding preference pair to be different for models capable of answering and those incapable. 2) on the other hand, existing LLMs such as Llama-2 initialize their reward models for RLHF from these generic preference pairs. Although subsequent optimization is based on human feedback and model sampling results, poor initialization can significantly affect the model's sampling space, making it difficult for the final model to have good reliability.
> > >
> > > [1] Joshi, Mandar, et al. "Triviaqa: A large scale distantly supervised challenge dataset for reading comprehension." arXiv preprint arXiv:1705.03551 (2017).
> > >
> > > [2] Bai, Yuntao, et al. "Training a helpful and harmless assistant with reinforcement learning from human feedback." arXiv preprint arXiv:2204.05862 (2022).
> > >
> > > [3] Cui, Ganqu, et al. "Ultrafeedback: Boosting language models with high-quality feedback." arXiv preprint arXiv:2310.01377 (2023).
> > >
> > > [4] Stiennon, Nisan, et al. "Learning to summarize with human feedback." Advances in Neural Information Processing Systems 33 (2020): 3008-3021.
> > >
> > > We hope the above responses can address your concerns.

---

> > ### Comment · Reviewer_tasN · 2024-06-06
> > **Questions**
> >
> > 1. Thank you for the new results on TriviaQA! How did you construct the preference data for the reward model? Moreover, this only compares RLKF to using a system prompt. Do you compare RLKF to regular RLHF?
> >
> > 2. It remains unclear to me why examples of "rejection > incorrect" preferences would not naturally be present in existing methods for preference data collection. However if these preferences are sampled from the LM, I buy the authors' argument that models do not naturally know how to decline unless explicitly taught.
> >
> > 3. Thank you for clarifying!
> >
> > 4. The statement that "*existing LLMs such as Llama-2 initialize their reward models for RLHF from these generic preference pairs*" is untrue, to the best of my understanding. I believe the reward model is initialized using responses sampled from (variants of) Llama2 (and continuously updated using the latest iteration of the finetuned model). It is not initialized from a static dataset of preference data.
> >
> > For reference, I share a snippet from the Llama 2 tech report below:
> > > We ask annotators to first write a prompt, then choose between **two sampled model responses**, based on provided criteria. In order to maximize the diversity, the two responses to a given prompt are sampled from **two different model variants**, and varying the temperature hyper-parameter.
> >
> > [...]
> >
> > > Human annotations were **collected in batches on a weekly basis**. As we collected more preference data, our reward models improved, and we were able to train progressively better versions for Llama 2-Chat (see the results in Section 5, Figure 20). Llama 2-Chat improvement also shifted the model’s data distribution. Since reward model accuracy can quickly degrade if not exposed to this new sample distribution, i.e., from hyper-specialization (Scialom et al., 2020b), **it is important before a new Llama 2-Chat tuning iteration to gather new preference data using the latest Llama 2-Chat iterations**. This step helps keep the reward model on-distribution and maintain an accurate reward for the latest model.

---

> ### Author Response · Authors · 2024-05-31
> **Comparison with Calibration-Based Methods**
>
> We further compare our method with calibration-based methods, and provide the results and our analysis below:
>
> |method|inference cost(dialogue turns * sampling num)|arithmetic(acc/truth/rely) |gsm8k(acc/truth/rely)|
> |---|---|---|---|
> |rlkf|1|31.9/88.1/56.5|17.0/59.6/41.5|
> |Raw logits(thresh-arith)[1]|1 |37.7/55.4/52.3|23.7/24.9/24.9|
> |Raw logits(thresh-gsm8k)[1]|1 |21.2/85.9/44.0|13.9/72.2/38.2|
> |P(True) [3]| 1|17.6/71.3/42.5|19.5/39.3/35.4|
> |verb. 1S top-1 [4]|1|16.7/50.2/39.0|4.8/11.6/11.1|
> |verb. 2S top-1 [4]|2|14.7/87.8/34.4|4.4/20.7/18.0|
> |agreement(consistency)[1]|10 |37.9/79.9/62.3|20.1/77.1/44.6|
>
> - **Unable to Reject Explicitly**: Calibration-based methods need to search and determine the best threshold for rejection and provide human-crafted rejection templates as responses. As we shown in the table, we search the threshold for arithmetic and gsm8k separately (on 100 validation cases from each dataset). However, the thresholds are quite different for different datasets which results in significant performance degradation with different thresholds.  In contrast, our method can enable the model to reject out-of-knowledge questions with personalized responses for different prompts automatically.
>
> - **High Inference Cost**: Consistency-based methods, on the one hand, require multiple samplings to obtain results, and on the other hand, may necessitate the use of additional models to extract answers for voting (we use ChatGPT to extract answers because rule-based methods may result in inaccurate extraction). This results in 5(sampling num) * 2(1 for answer generation, 1 for answer extraction) = 10 times (or at least 5 times) the inference cost than other methods. Some Verbalized-based methods (verb. 2S) also require the model to generate confidence through an additional round of response after generating the answer.
>
> - **High Calibration Variance**: Utilizing calibration methods to determine the accuracy of answers is not stable. For instance, logit-based methods are not quite reasonable when the model generates longer responses, and Verbalized-based methods result in significant fluctuations in confidence scores and even prediction results (as shown in the gsm8k results of verb. methods) due to the variability in prompts.
>
> In summary,  calibration methods are more suitable for analyzing the uncertainty of model responses or constructing training data (such as the self-consistency[5] introduced in our paper). However, our alignment research on reliability aims to enable the model to acquire self-knowledge and explicitly refuse out-of-knowledge questions. Experimental results show our method can enable the model to reject automatically without additional inference costs and improve the accuracy of rejections compared to most calibration methods.
>
> [1] Calibrating Large Language Models with Sample Consistency
>
> [2] Calibration-Tuning: Teaching Large Language Models to Know What They Don’t Know
>
> [3] Language Models (Mostly) Know What They Know
>
> [4] Just Ask for Calibration: Strategies for Eliciting Calibrated Confidence Scores from Language Models Fine-Tuned with Human Feedback
>
> [5] Self-consistency improves chain of thought reasoning in language models.

---

> ### Comment · Area_Chair_kdZ7 · 2024-06-04
>
> Hi tasN, can you check the authors' response and update your review if it addressed your concern (or participate in discussion with the authors if it did not)?

---

> ### Author Response · Authors · 2024-06-07
>
> Thanks for your comments! We give our responses below:
>
> # 1. Re: Comment 1
>
> - We construct the preference pairs following the same construction setting of in-domain preference data in our paper.  When the model's multiple sampling results are all correct, we use pairs where the correct answer is preferred over a refusal. When all results are incorrect, we use pairs where a refusal is preferred over the incorrect answer. When the sampling results are mixed, we use pairs where the correct answer is preferred over the incorrect answer. We use the exact match metric instead of accuracy to determine if each sampling result is correct.
> - Due to time constraints, we did not run the RLHF baseline. We will include the relevant experiments in future version. Additionally, since the reward model used in RLHF does not incorporate the preference data regarding refusals as in RLKF, we believe it cannot effectively improve the model's truthfulness, as demonstrated by our results on the arithmetic dataset in the paper.
>
> # 2. Re: Comment 4
>
> Thank you for your clarification! We indeed had a misunderstanding regarding the initialization of the reward model in Llama-2. We will correct the inaccurate claims in our paper. The core difference between our method and the RLHF method is that we explicitly introduced the rejection preference pairs in the training of the reward model. This allows the reward model to better understand when to refuse. As shown in Table 4 of our paper, the reward model trained with helpful preference data only achieved an accuracy of 49.9 on the "beyond" subset of the in-domain arithmetic reliable preference data (this task is binary classification, where the reward model chooses between refusing to respond and responding incorrectly, with the correct label being refusal). However, after adding the corresponding in-domain reliable preference data (i.e., various preference pairs regarding refusals), this classification accuracy significantly improved to 83.0. This partly explains why RLKF can better improve the model's reliability compared to RLHF, as the reward model after RLKF can better determine when it should refuse.

---

### Comment · Area_Chair_kdZ7 · 2024-06-02
**Discussion period is now open**

Hi reviewers, please take a look at the author's rebuttals and the other reviews for this paper!

If the rebuttals addressed your concerns, please let the authors know about this and update your review. If not, please continue to engage with the authors and the other reviewers in the discussion forum.

Reviewer tasN, you have concerns about missing related work, a limited scope of experiments relying on synthetic benchmarks, and inaccuracies / opacity in the writing. Does the author's response address these concerns?

qwqv and TGqk are both positive about this paper's acceptance. Would you like to argue for its acceptance, or do you share concerns with the other reviewers?

---

### Decision · Program_Chairs · 2024-07-10

**Decision:**

Accept

**Comment:**

This paper addresses the problem of model hallucination, when models generate incorrect responses to questions. The paper proposes that a fundamental underlying problem is that models cannot recognize when they cannot produce a correct answer, and that one solution is to train models to explicitly reject questions when they are not able to answer correctly. Authors propose a RL-based fine-tuning method that uses external knowledge and/or self-consistency to create train-time preference data that incentivizes correctness and consistency of outputs.

Reviewers point out several limitations of the paper. Experiments are limited by the choice of benchmarks (arithmetic only, though authors add TriviaQA results in the rebuttal, though not with direct comparison to RLHF), and comparison / engagement with methods in calibrating language models is missing. The rebuttal contains additional experiments comparing with other calibration-based methods, showing mostly mixed results.

[comment from PCs] Please incorporate the additional experiments into the paper, even if results are mixed, to provide a complete picture of the approach.